# Bibliometric analysis of hotspots and frontiers in cancer-related fatigue among ovarian cancer survivors

Yuanxia Liu[1], Qianxia Liu[2], Xiaolian Jiang [1]*

1 West China School of Nursing, Sichuan University/ West China Hospital, Sichuan University, Sichuan, China, 2 Department of Laboratory, Linxia Hui Autonomous Prefecture Center for Disease Control and Prevention, Gansu, China

* jiangxiaolianhl@163.com

## Abstract

### Objectives

To explore and analyze research hotspots and frontiers in CRF in ovarian cancer patients to provide an evidence-based basis for scholars and policymakers.

### Background

Ovarian cancer is one of the most common and lethal gynecological malignancies. Cancer-related fatigue (CRF) is an annoying and pervasive side-effect that seriously affects the activities of daily living and decreases the quality of life (QoL) of cancer survivors.

### Methods

The literature was retrieved from the Web of Science Core Collection (WOSCC) from inception to 2021-12-31. CiteSpace was used to discuss research countries, institutions, authors, and keywords.

### Results

This study ultimately included 755 valid publications, and the number of publications showed a gradual upward trend. The countries, institutions, authors, and journals that have published the most articles and cited the most frequently were the United States, the University of Texas MD Anderson Cancer Center, Michael Friedlander and Amit M Oza, Gynecologic Oncology, and Journal of Clinical Oncology. The top three high-frequency keywords were Ovarian cancer, chemotherapy, and clinical trial. The top three keywords with the strongest citation bursts were cyclophosphamide, double-blind, and open-label.

### Conclusions

Conducting multi-center, large-sample, randomized controlled clinical trials to determine whether chemotherapeutic agents have severe adverse effects and to discuss the relationship between CRF and QoL and overall survival in cancer survivors are hotspots in this field.

**Data Availability Statement:** All relevant data are within the paper and its Supporting Information files.

**Funding:** The author(s) received no specific funding for this work.

**Competing interests:** The authors have declared that no competing interests exist.

The new trends may be applying double-blind, randomized controlled trials to clarify the causes of CRF and open-label, randomized trials to determine the efficacy, safety, and tolerability of chemotherapeutic agents.

## Background

Ovarian cancer is one of the most common and lethal gynecological malignancies, accounting for approximately 4% of global cancer incidence and mortality in women [1, 2]. Due to its unique anatomical site and generally vague symptoms, even in high-income countries (HICs), more than 60% of ovarian cancer are detected only in the late stage [3, 4]. Surveys conducted by the Global Burden of Diseases Cancer Collaboration note that incident cases of ovarian cancer worldwide increased from 286,000 to 294,000, and cancer deaths increased from 176,000 to 198,000 from 2017 to 2019 [5, 6]. Fortunately, with better diagnostic techniques, more and more ovarian cancer is being detected at an early stage, and early detection can significantly improve the health outcome of patients with ovarian cancer and save their lives [7]. We can learn from the latest studies that the survival rate of ovarian cancer after five years ranges from 46.2% [7] to 47.4% [3]. The absolute number of global ovarian cancer-related disability-adjusted life years (DALYs) increased by 27.5% from 2010 to 2019, reaching 5.36 million in 2019 [6].

Cancer-related fatigue (CRF) is an annoying and pervasive side-effect of cancer and cancer therapy [8, 9]. According to the definition developed by the National Comprehensive Cancer Network (NCCN), CRF is a type of fatigue that persists and cannot be relieved by rest [10]. The reported incidence of CRF varies due to the different scales chosen to assess symptoms. Studies have shown that the incidence ranges from 25% to 99% [11], with a 90% incidence in patients undergoing radiotherapy and 80% in those undergoing chemotherapy [12], and the pooled prevalence of CRF was 52% [13]. A survey showed that CRF persists for several years even after treatment ends for 25%-33% of cancer survivors [14]. CRF has adverse effects on various physiological, psychological, and social aspects and severely diminishes the QoL of cancer survivors [15]. Several studies have shown that ovarian cancer patients with CRF had poor QoL [16–18]. Relieving CRF in ovarian cancer survivors could improve their self-care ability and enhance QoL.

Bibliometric analysis is a scientific mathematical and statistical method to quantitatively explore the development of a research field, and knowledge mapping is a visual representation of bibliometric analysis [19]. The method can outline the development of a specific research field or topic from a macro perspective, clarify its research hotspots, detect its research frontiers, and provide a reference basis for the future research direction of the field or topic [20]. CiteSpace is an effective visualization tool that is widely used for knowledge mapping in the fields of medicine [21], informatics [22], and science [23].

Currently, a bibliometric analysis of CRF in people with ovarian cancer has not been published. The purpose of this study was to analyze the literature related to CRF among ovarian cancer patients published in WOSCC from 1991 to 2021 by using the visualization tool CiteSpace. To explore and analyze research hotspots and frontiers in CRF in patients with ovarian cancer to provide an evidence-based basis for scholars and policymakers.

## Materials and methods

### Data source

Selecting the proper database and searching thoroughly and accurately for literature related to the topic is the first step of research. The Web of Science was chosen as the data source for the

following reasons. Firstly, the Web of Science is an international multidisciplinary citation index publication on citation statistics and is recognized worldwide as the most authoritative indexing tool for scientific and technical literature, consisting of the WOSCC and other databases [24, 25]. Secondly, the Web of Science is also the largest and most comprehensive resource for academic information covering a wide range of disciplines, with data updated to reflect dynamic changes in the scientific field timely [26]. Moreover, the Web of Science has the advantage of being a bibliographic database that provides references cited in scholarly publications [27].

## Data processing

The WOSCC was searched for investigating literature on CRF in women with ovarian cancer from inception to December 31st, 2021. The first article on CRF in ovarian cancer patients in the WOSCC was issued in 1991, and the literature search date was December 31, 2021; therefore, the timespan of our study was 1991.01.01–2021.12.31. The retrieval subject terms are ovarian cancer and cancer-related fatigue. A total of 768 original records were retrieved. The retrieved documents were imported into CiteSpace to remove duplicates and then independently screened by 2 researchers to exclude papers that did not fit the research topic and search strategy. After excluding editorial materials (n = 3), meeting abstracts (n = 8), and proceedings papers(n = 2), 693 articles and 62 reviews related to the topic were finally obtained.

## Data analysis

CiteSpace is a Java application and bibliometric software developed by Professor Chaomei Chen [28]. CiteSpace applies co-citation analysis theory and pathfinder network algorithms to extract information from scientific literature, converting the information into visualized mapping knowledge maps and presenting the research evolution, hotspots, and frontiers of a given area [29, 30].

In this study, CiteSpaceV5.8 R3 visualization software was used for keyword extraction, keyword burst detection, and clustering analysis. Keyword burst detection and keyword clustering analysis were performed using the g-index algorithm. Each node in the map represents a keyword, and a line between two nodes indicates a connection between two nodes [31]. The larger the node, the more frequently the keyword appears. Node centrality represents the connection between a node and other nodes, as well as the position and role of a node in the whole network [30, 32]. If the centrality of a node is higher than 0.1, it means that the node has a relatively core position in the network [33]. In the timeline diagram, the red part of the green timeline represents the starting and ending years of the cited keywords, and "strength" represents the strength of the cited keywords [34].

## Results

### Analysis of annual publications

The volume of scientific literature publications could reflect the dynamic process of research ups and downs and reflect the level of scientific research in a particular field [35]. With 755 sorted and detected papers as samples, the publication volume of literature in the 31 years from 1991 to 2021 was statistically analyzed in the time dimension, mapping out the chronological distribution of relevant studies (Fig 1).

Three research articles in this area were published in 1991. Sandra et al. conducted a small-scale (twelve ovarian cancer patients in the test group and twelve healthy adult women in the control group) descriptive study indicating no clear link between fatigue and diseases or

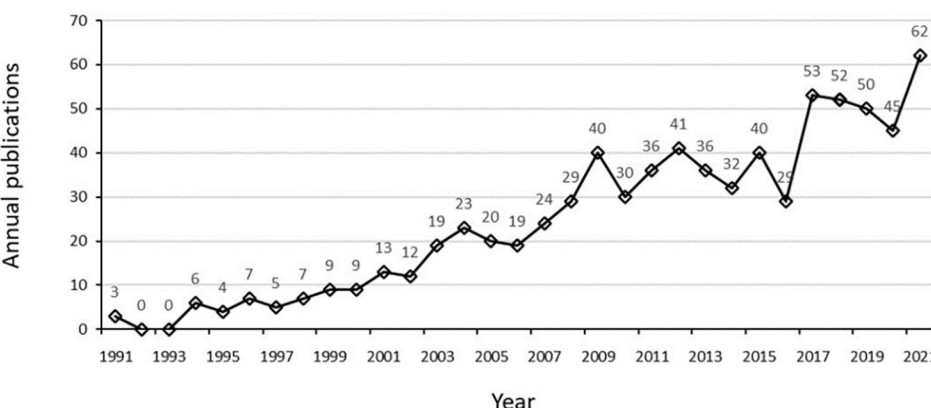

**Fig 1. Annual publications on cancer-related fatigue in ovarian cancer survivors.**

treatments in ovarian cancer survivors undergoing chemotherapy [36]. P. J. Disaia et al. found that concurrent use of chemotherapeutic agents and interferon in patients with ovarian cancer led to severe fatigue [37]. Michele A. Hood and Rebecca S. Finley did not specify whether having cancer or chemotherapy affected fatigue in ovarian cancer patients treated with fludarabine [38]. Since then, research on this theme has begun to sprout.

## Performance of countries /regions

The country was set as node type and time slice was set as 1 year, and 755 records of CRF in ovarian cancer patients published from 1991 to 2021 were analyzed to generate a collaborative country network map with 61 nodes, 375 links, and a density of 0.2049(Fig 2). Each node in the figure represents a country (or a region), and the size of the node is proportional to the papers published by countries [39]. The links between nodes represent cooperation between countries, and the thickness of the links is positively correlated with the number of articles issued.

The top 10 active countries in this research field (n> = 40) are listed in Table 1. Combining Fig 2 and Table 1 and counting the total number of publications in CiteSpace showed that 1,236 papers were published in 61 countries, with 938 papers published in the ten most active countries, far exceeding the total literature related to the topic. It showed strong links between research countries and frequent cooperation between researchers in this field. The top 10 countries accounted for 75.9% of the total publications, while the United States had the most publications with 426, accounting for 34.5% of the total publications. The United States, Australia, Italy, and France have a centrality of over 0.1.

## Performance of institutions

With the institution as the node type and time slice of 1 year, 755 records of CRF in ovarian cancer survivors were analyzed to generate a knowledge map (Fig 3). Pathfinder and pruning sliced networks were used to present a more intuitive and understandable map. 634 institutions studied CRF in ovarian cancer patients. Each node in Fig 3 represents an institution, and the size of the node is proportional to the papers published by institutions. The links between nodes represent cooperation between institutions, and the thickness of the links is positively correlated with the number of articles issued.

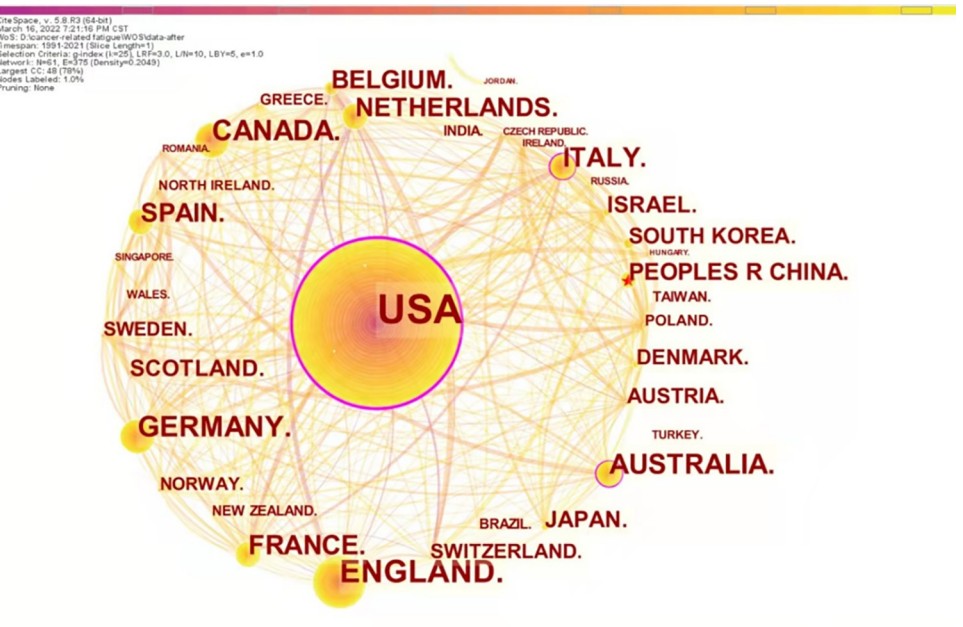

**Fig 2. Map of research countries in cancer-related fatigue in ovarian cancer survivors.**

The top 10 productive institutions are listed in Table 1. Combining Fig 3 and Table 1 and counting the total number of publications in CiteSpace, 634 institutions published 1512 papers, far exceeding the total literature related to the theme. Among these research institutions with more than 20 publications were the University of Texas MD Anderson Cancer Center, Dana-Farber Cancer Institute, The University of Oklahoma, Memorial Sloan-Kettering Cancer Center, Massachusetts General Hospital, and National Cancer Institute (NCI).

## Analysis of authors

The author was set as node type, and years per slice was 1 year and generated a collaborative author map with 798 nodes, 1369 links, and a density of 0.0043 (Fig 4). As seen in Fig 4, the nodes represent the researchers in the field, and the node size is proportional to the papers published by researchers. The links between nodes reflect the collaboration between authors, and the thickness of the links is proportional to the outputs. Core authors refer to researchers

**Table 1. Top 10 research countries and institutions.**

| Ranking | Countries | | | Institutions | | |
|---|---|---|---|---|---|---|
| | Countries | Publications | Centrality | Institutions | Publications | Centrality |
| 1 | USA | 426 | 0.24 | The University of Texas MD Anderson Cancer Center | 56 | 0.02 |
| 2 | England | 93 | 0.06 | Dana-Farber Cancer Institute | 47 | 0.13 |
| 3 | Canada | 66 | 0.09 | The University of Oklahoma | 34 | 0.02 |
| 4 | Germany | 64 | 0.09 | Memorial Sloan-Kettering Cancer Center | 30 | 0.05 |
| 5 | Australia | 55 | 0.15 | Massachusetts General Hospital | 29 | 0.04 |
| 6 | Italy | 51 | 0.12 | National Cancer Institute (NCI) | 22 | 0.02 |
| 7 | France | 50 | 0.10 | The Ohio State University | 19 | 0.00 |
| 8 | Netherlands | 48 | 0.04 | Leon Berard Center | 18 | 0.11 |
| 9 | Spain | 44 | 0.03 | Texas State University | 17 | 0.05 |
| 10 | Belgium | 41 | 0.06 | Roswell Park Cancer Center | 16 | 0.05 |

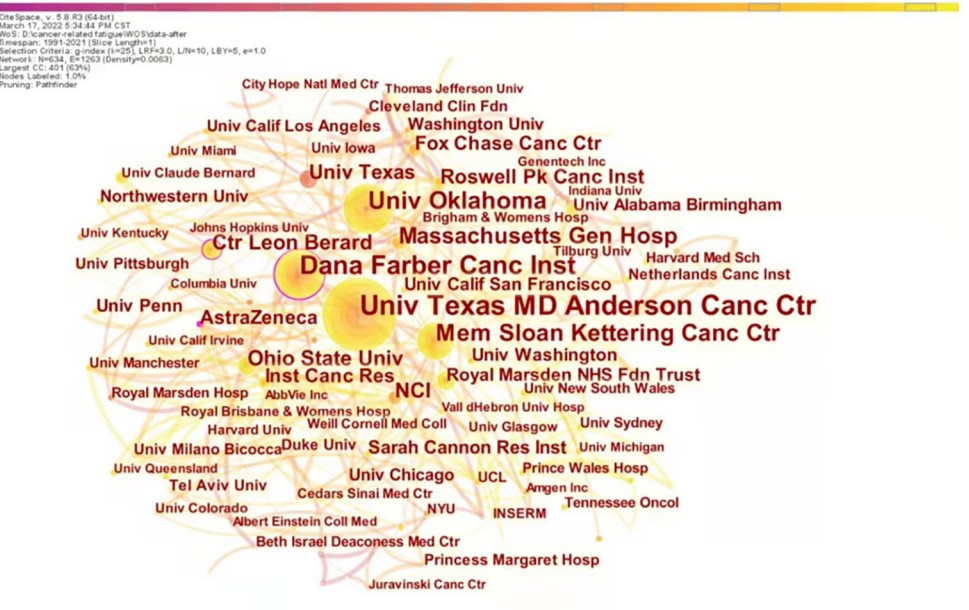

**Fig 3. Map of research institutions in cancer-related fatigue in ovarian cancer survivors.**

with high academic attainments, extensive publications, and profound influence in a particular field. Price's Law [40] defines core authors by the number of papers published, and its calculation formula is M = 0.749* Nmax1/2 (Nmax refers to the number of papers published by the most productive author in the statistical years, and those who published more than M papers are core authors of this research area). The most productive author is Michael Friedlander and Amit M Oza, with 17 publications. According to the calculation formula, Nmax and M for this study area were 17 and 6.4, respectively. Therefore, authors with publications of 7 or more were the core authors. The core authors in this domain are listed in Table 2. Combined to Fig 4 and Table 2, 798 authors published 755 papers, while 119 papers were published by the 11 core authors, accounting for 15.8% of the total publications in the field.

## Analysis of journals and cited-journals

Since the first article on CRF in women with ovarian cancer was published in 1991, the overall trend of published articles in this area has been increasing year by year. Papers on ovarian cancer women with CRF were published in approximately 200 journals till 2021. An analysis of cited journals in CiteSpace shows that 755 papers were cited in 722 journals. Gynecologic Oncology ranked first among all journals publishing studies related to cancer fatigue in ovarian cancer patients with 142 publications and second among all co-cited journals with a frequency of 640. Gynecologic Oncology is dedicated to publishing clinical and investigative articles on female genital tract tumors. Its impact factor (IF) is 5.482, and the average number of citations per paper in this journal is 7.4. Journal of Clinical Oncology ranked second among all journals with 142 publications and second among all co-cited journals with a frequency of 457, and focused on treating various types of malignant disease. Its impact factor is 44.544, and the average number of citations per paper in this journal is 37. Articles published in Gynecologic Oncology and Journal of Clinical Oncology accounted for 25.4% of the total publications in this area. The top 10 productive journals and co-cited journals related to CRF in women with ovarian cancer are shown in Table 3.

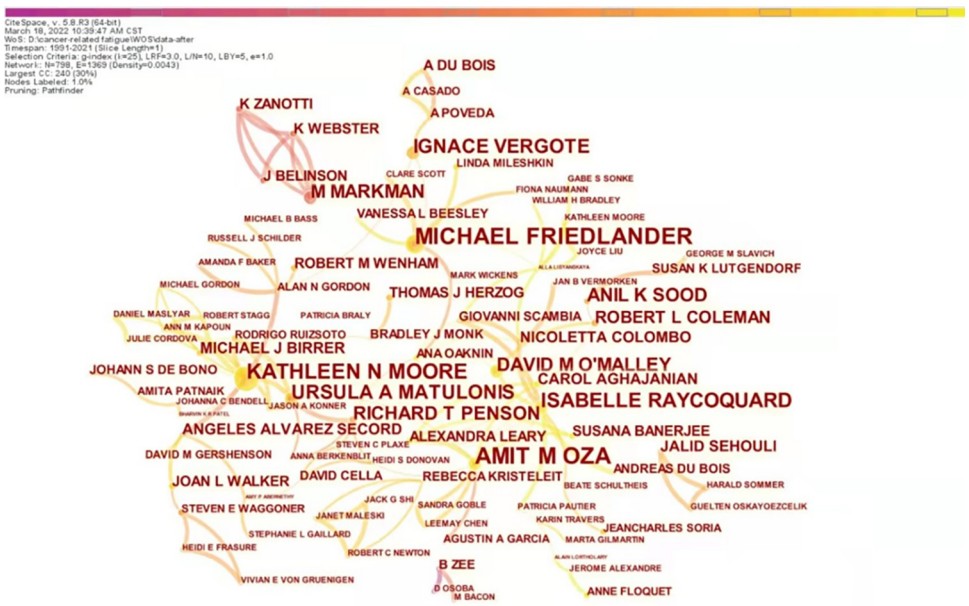

**Fig 4. Map of author network related to cancer-related fatigue in ovarian cancer survivors.**

**Table 2. Core authors of researching CRF in ovarian cancer survivors.**

| Ranking | Authors | Counts | Centrality |
|---|---|---|---|
| 1 | Michael Friedlander | 17 | 0.06 |
| 2 | Amit M Oza | 17 | 0.06 |
| 3 | Kathleen N Moore | 15 | 0.03 |
| 4 | Isabelle Ray-Coquard | 12 | 0.04 |
| 5 | Ignace Vergote | 10 | 0.05 |
| 6 | Ursula A Matulonis | 10 | 0.03 |
| 7 | David M O'Malley | 9 | 0.01 |
| 8 | Richard T Penson | 9 | 0.01 |
| 9 | M Markman | 9 | 0.00 |
| 10 | Anil K Soon | 9 | 0.02 |
| 11 | Robert L Coleman | 7 | 0.02 |

**Table 3. Top 10 most productive journals and co-cited journals.**

| Raking | Journals | IF (2021) | Publications | Co-cited-journal | IF (2021) | Frequency |
|---|---|---|---|---|---|---|
| 1 | Gynecologic Oncology | 5.482 | 142 | Journal of Clinical Oncology | 44.544 | 640 |
| 2 | Journal of Clinical Oncology | 44.544 | 50 | Gynecologic Oncology | 5.482 | 457 |
| 3 | Clinical Cancer Research | 9.619 | 45 | Annals of Oncology | 32.976 | 337 |
| 4 | Investigational New Drugs | 3.850 | 31 | New England Journal of Medicine | 91.245 | 335 |
| 5 | International Journal of Gynecological Cancer | 3.437 | 30 | Clinical Cancer Research | 9.619 | 299 |
| 6 | Annals of Oncology | 32.976 | 29 | European Journal of Cancer | 9.162 | 295 |
| 7 | Cancer | 6.860 | 24 | British Journal of Cancer | 7.640 | 275 |
| 8 | British Journal of Cancer | 7.640 | 20 | Cancer Research | 12.701 | 273 |
| 9 | Cancer Chemotherapy and Pharmacology | 3.333 | 18 | Cancer | 6.860 | 251 |
| 10 | Supportive Care in Cancer | 3.603 | 14 | Journal of the national cancer institute | 13.506 | 251 |

**Table 4. The top 10 keywords in terms of counts and centrality.**

| Ranking | Keywords | Counts | Centrality |
|---|---|---|---|
| 1 | Ovarian cancer | 522 | 0.08 |
| 2 | Chemotherapy | 222 | 0.08 |
| 3 | Clinical trial | 166 | 0.14 |
| 4 | Quality of life | 140 | 0.13 |
| 5 | Breast cancer | 102 | 0.18 |
| 6 | Cisplatin | 78 | 0.09 |
| 7 | Paclitaxel | 77 | 0.06 |
| 8 | Cancer-related fatigue | 65 | 0.06 |
| 9 | Cancer survivor | 57 | 0.15 |
| 10 | Bevacizumab | 52 | 0.04 |

## Research hotspots and frontiers on CRF in ovarian cancer survivors

Keywords could be accurately extracted throughout the paper, and their frequency is a kind of mapping of the research hotspots. CiteSpace can identify the frontier of a certain research field by detecting burst words with high frequency and fast growth. This section used the keyword as node type to analyze the main research hotspots and frontiers from keyword co-occurrence and clustering and keyword bursts. To determine research hotspots and frontiers in this area, we extracted the top 10 keywords from 755 articles and listed them in Table 4.

## Keyword co-occurrence and clustering

688 keywords were extracted from 755 relevant articles, as shown in (Fig 5). Nodes represent keywords in this area, and links represent the relationship between nodes. The larger the node and the thicker the line, the more crucial the node is.

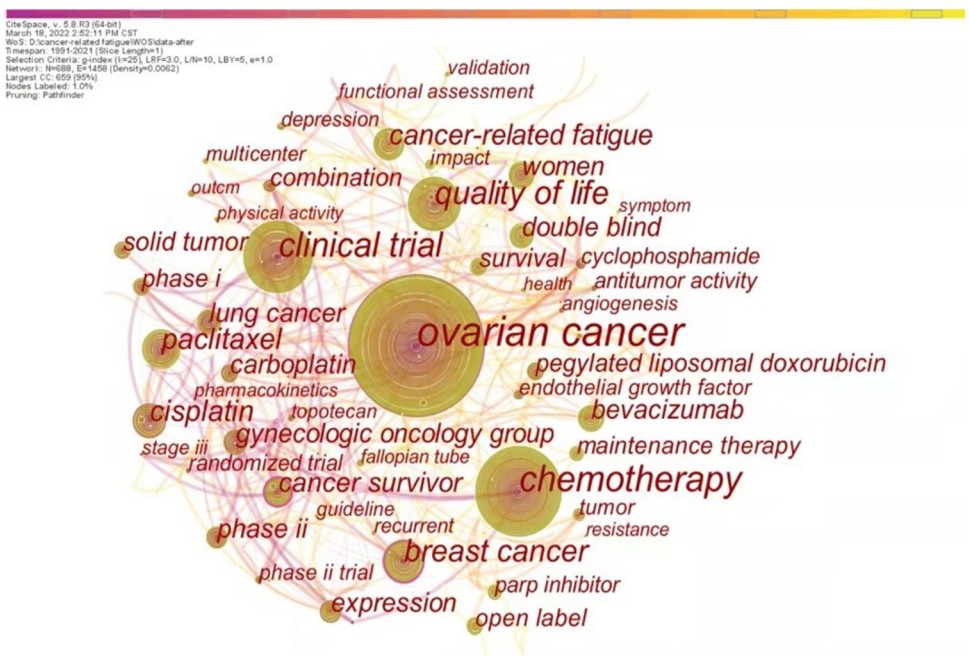

**Fig 5. Map of keyword co-occurrence network in cancer-related fatigue in ovarian cancer survivors.**

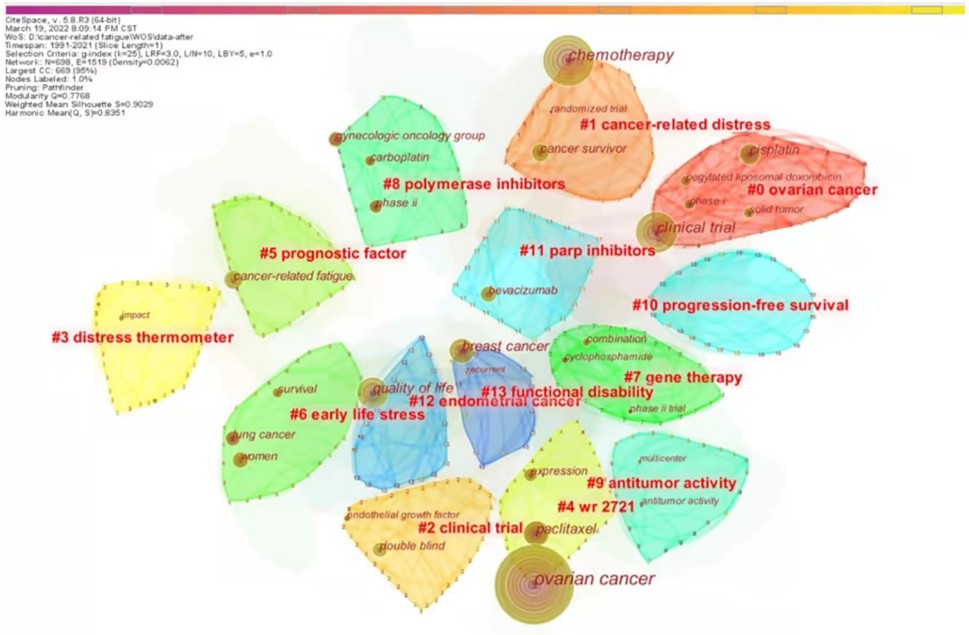

**Fig 6. Map of keyword clustering in cancer-related fatigue in ovarian cancer survivors.**

As shown in Table 4, the subject terms of this study are cancer-related fatigue and ovarian cancer, ranking in the 8th and 1st of the top 10 keywords, respectively. Other the most frequent keywords related to our study are chemotherapy, clinical trial, quality of life, and cancer survivor. The clinical trial, QoL, and cancer survivor have high centrality (centrality $\geq$ = 0.1).

In this study, the LSI algorithm recommended by Professor Chaomei Chen was applied to cluster the keywords, and the keyword clustering knowledge map is generated after adjustment and filtering, as shown in (Fig 6). According to the clarity of the network structure and clustering, CiteSpace provides the modularity (Q) index and average silhouette score to evaluate the drawing effect of the map. The average silhouette score and Q score are both between -1 and 1, and the higher the average silhouette score, the more reliable the clustering is; the higher the Q score, the better the structure of the network [41, 42]. The Q score of this study is 0.7768, and the average silhouette score is 0.9029, which can be considered that this clustering is significant and convincing.

Fig 6 shows 14 clustering modules with different color blocks to distinguish each cluster. The 14 clustered modules are labeled with #0-#13, the outline of the module is marked with the corresponding number, and the core keywords of the module are displayed within each module. The keyword font size and the circle size in front of them are proportional to the frequency of the keyword. The cluster structure and core keywords in Fig 6 reflect some of the research hotspots in CRF in ovarian cancer patients. Mainly, cluster 0 is ovarian cancer and the core keyword is clinical trial. Cluster 1 is cancer-related distress, and the core keywords are chemotherapy and cancer survivor. Cluster 4 is wr2721 with core keywords ovarian cancer and paclitaxel. Cluster 5 is prognostic factors, and the core keyword is CRF. Cluster 11 is parp inhibitors, and the core keyword is bevacizumab. Cluster 12 is endometrial cancer, and the core keyword is quality of life. Cluster 13 is functional disability, and the core keyword was breast cancer.

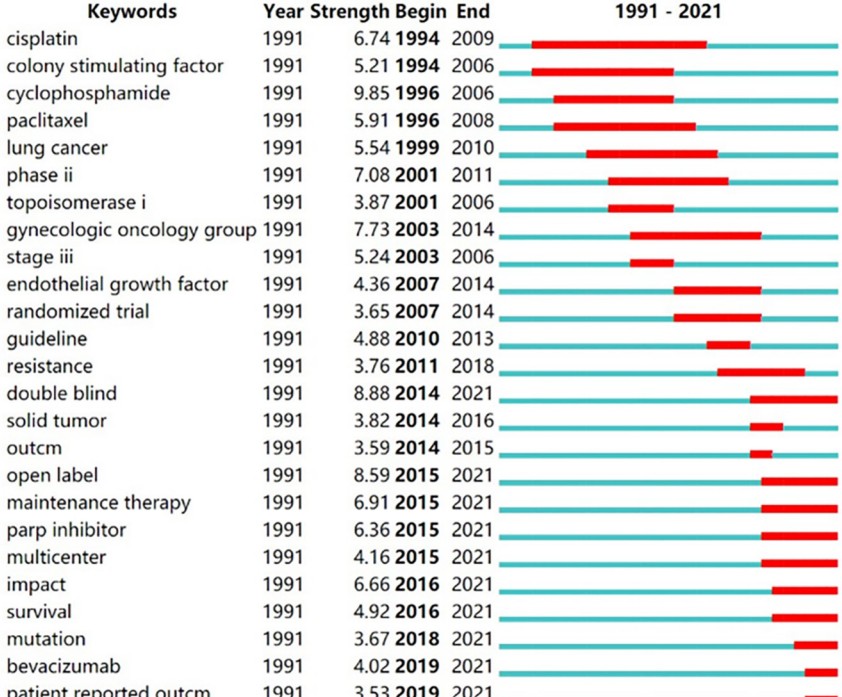

**Fig 7. Map of the top 25 keywords with the strongest citation bursts.**

## Keyword bursts

"Burst words" are keywords that suddenly increase in citation frequency within a certain period, revealing the research hotspots and frontiers in the area [43–45]. The top 25 strongest burst words and the year of begin and end of each burst word in the field of CRF in ovarian cancer survivors as shown in (Fig 7). The green line in Fig 7 displays the time window of the documents retrieved in this study, and the red color indicates the duration of a certain burst word from its beginning to its end. The first burst keywords were cisplatin and colony stimulating factor in 1994, and the keywords with the highest burst strength were cyclophosphamide, double-blind, and open-label.

## Discussion

Based on the WOSCC, this study applied CiteSpace to explore the research status of CRF in ovarian cancer patients, and analyze research hotspots and frontiers by analyzing 755 papers over 31 years.

### The current research status for CRF in ovarian cancer survivors

The current state of research is described in terms of annual publications, research countries, institutions, researchers, journals, and cited journals [46]. As can be seen from the annual publications in this field, there has been no research in this area for the next two years since the first paper was published in 1991. Over the next decade, research in this area showed a slow-growth trend, with consistently fewer than ten papers published per year. In the second

decade, annual publications increased significantly compared to the previous decade, with a slight decline in 2010. The volume of research literature had fluctuated considerably in the last decade, with an overall upward trend, reaching a new peak in 2021 when the annual publications increased to 62. Based on the performance of the study countries and institutions, the United States took an absolute leadership role in CRF in ovarian cancer people research with high publications (count = 426) and high centrality (centrality = 0.24), the University of Texas MD Anderson Cancer ranked first with 56 publications. Dana-Farber Cancer Institute is the most central institution for research in the area, with the highest centrality of 0.13, indicating that their literature strongly influences this research area. An analysis of researchers in this field shows 11 core authors, of which those with at least ten publications are Michael Friedlander, Amit M Oza, Kathleen N Moore, Isabelle Ray-Coquard, Ignace Vergote, and Ursula A Matulonis. Collaboration contributes to professional development and the exploration of research frontiers and should be sought to the maximum extent possible between countries, institutions, and researchers.

Papers on CRF in ovarian cancer patients have been published in approximately 200 journals, and 39.5% of the papers were published in the top five journals. The two journals with the highest publications are Gynecologic Oncology and Journal of Clinical Oncology. 62 papers were published in 45 journals in 2021, 15 of which were published in gynecologic oncology. Lancet Oncology, the highest impact factor among these 45 journals, which published a study of a randomized controlled trial showed that the proportion of patients with ovarian cancer accompanied with grade 3 fatigue in avelumab combination with pegylated liposomal doxorubicin (PLD) group and PLD alone group was 10 (5%) and 3 (2%), respectively [47]. It indicates that CRF is trigging increasing attention as an adverse reaction during cancer and anti-tumor treatments.

## Research hotpots for CRF in ovarian cancer survivors

The co-occurrence network of high-frequency keywords represents a hot research topic in CRF in patients with ovarian cancer from 1991 to 2021. Since the search was conducted with ovarian cancer and CRF as the subject terms, although they appeared highest in the keyword co-occurrence knowledge map, it was not the focus of this paper; therefore, this was eliminated to obtain keywords with higher centrality and closer relevance to this study: chemotherapy, clinical trial, quality of life, cisplatin, paclitaxel, cancer survivor, and bevacizumab. As shown in Table 4, published articles on chemotherapy, clinical trial, and QoL were more than 100. In contrast, cancer survivor and the research on chemotherapeutic agents such as cisplatin, paclitaxel, and bevacizumab on fatigue are less studied and less published. The level of centrality is also a measure of the substantive impact that studying the hotspot can bring. The top 10 keywords with centrality greater than 0.1% were clinical trial, QoL, and cancer survivor.

Several surveys found that 43% [48]-53% [17] of patients with ovarian cancer complained of fatigue or extreme fatigue. CRF as a long-term side effect in cancer survivors negatively impacts the QoL of patients [18, 49]. Home-based exercise, good relationships, and lifestyle interventions can improve QoL and reduce fatigue in patients with ovarian cancer [50, 51]. Moreover, a randomized controlled trial study indicated that QoL was associated with overall survival in ovarian cancer patients [52]. In addition, studies have shown that chemotherapeutic agents such as cisplatin, paclitaxel, carboplatin, bevacizumab, and PARP inhibitors can also trigger fatigue in patients, and the percentage of fatigue caused ranges from 28%-35% [53–55]. Therefore, conducting multi-center, large-sample, randomized controlled clinical trials to determine whether chemotherapeutic agents have severe adverse effects in patients with ovarian cancer and to discuss the relationship between CRF and QoL and overall survival in cancer survivors are hotspots in this field.

## Analysis of frontiers for CRF in ovarian cancer survivors

Analyzing keyword bursts could help researchers quickly understand the frontier or future trends in a particular field [56]. The highest strength burst keyword was cyclophosphamide with a strength of 9.85, which began in 1996 and ended in 2006. The research literature on cyclophosphamide in this period explored the effects of chemotherapeutic drug application on fatigue in patients with ovarian cancer. During the combination of different chemotherapeutic agents with cyclophosphamide, patients experienced fatigue grades ranging from grade 1 to 3, and the probability of occurrence varied from 8.3% [57] to 66% [58].

Double-blind ranked second with a strength of 8.88, which began in 2014 and persists till now. Double-blind is a vital principle in the design of randomized controlled trials, which can decrease bias due to subjective factors of subjects and researchers, although there are difficulties in its implementation [59]. We can learn from a randomized, double-blind study that ovarian cancer patients treated with two chemotherapeutic agents were more susceptible to fatigue than those treated with only one (87.7% vs. 74.1%) [60]. In another double-blind, placebo-controlled trial, the ratio of experiencing grade 1–2 fatigue and grade 3 fatigue in the trial group of patients with ovarian cancer was 62% and 4%, and the ratio of experiencing grade 1–2 fatigue and grade 3 fatigue in the placebo group was 37% and 2%, indicating that cancer itself is also a major contributor to fatigue [59].

Open-label ranked third with the strength of 8.59, which began in 2015 and continues until now. An open-label trial is a clinical trial in which both the investigators and participants know the drug or treatment being given [61]. Researchers have found that open-label placebo had a significant effect on some subjective symptoms [62]. In recent years, the application of open-label trials to determine the efficacy, safety, and tolerability of chemotherapeutic agents has become a new trend in scientific research [47, 63–65].

## Strengths and limitations

To our knowledge, this is the first study to use the co-occurrence and co-citation analysis methods by CiteSpace to perform bibliometric analysis and visual display of CRF among ovarian cancer patients. Moreover, we deeply discuss the research status and explore the hotspots and frontiers in this field. There are also several limitations to this study. Due to the limitation of the bibliometric software, this study only retrieved data from the WOSCC, failing to include all research literature in the field of CRF in ovarian cancer survivors, which may be biased owing to insufficient research data. A comprehensive review of the literature to determine research hotspots and frontiers is an aspect that needs to be improved in future research.

## Conclusions

In conclusion, research hotspots and frontiers in studies related to CRF in ovarian cancer patients in the past 31 years are always in dynamic change and difficult to precisely control. This review analyzes the research hotspots and frontiers in this area based on the existing literature to provide an evidence-based basis for policymakers and researchers. Conducting multi-center, large-sample, randomized controlled clinical trials to determine whether chemotherapeutic agents have severe adverse effects in ovarian cancer patients and to discuss the relationship between CRF and QoL and overall survival in cancer survivors are hotspots in this field. The new trends may be applying double-blind, randomized controlled trials to clarify the causes of cancer-related fatigue and open-label, randomized trials to determine the efficacy, safety, and tolerability of chemotherapeutic agents.

## Acknowledgments

We would like to express our gratitude to those who provided help in data collecting and paper writing.

## Author Contributions

**Conceptualization:** Yuanxia Liu.

**Data curation:** Yuanxia Liu, Qianxia Liu.

**Formal analysis:** Yuanxia Liu.

**Investigation:** Yuanxia Liu.

**Methodology:** Yuanxia Liu, Qianxia Liu, Xiaolian Jiang.

**Project administration:** Xiaolian Jiang.

**Software:** Yuanxia Liu.

**Supervision:** Xiaolian Jiang.

**Validation:** Yuanxia Liu, Qianxia Liu, Xiaolian Jiang.

**Visualization:** Yuanxia Liu.

**Writing – original draft:** Yuanxia Liu.

**Writing – review & editing:** Yuanxia Liu, Qianxia Liu.

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
