## [Decision Letter · Decision Letter 0]

8 Aug 2022

PONE-D-22-12589Bibliometric Analysis of Hotspots and Frontiers in Cancer-Related Fatigue Among Ovarian Cancer SurvivorsPLOS ONE

Dear Dr. Jiang,

Thank you for submitting your manuscript to PLOS ONE. After careful consideration, we feel that it has merit but does not fully meet PLOS ONE’s publication criteria as it currently stands. Therefore, we invite you to submit a revised version of the manuscript that addresses the points raised during the review process.

We look forward to receiving your revised manuscript.

Kind regards,

Kehinde Sharafadeen Okunade

Academic Editor

PLOS ONE

Journal Requirements:

Reviewers' comments:

Reviewer's Responses to Questions

**Comments to the Author**

1. Is the manuscript technically sound, and do the data support the conclusions?

Reviewer #1: Yes

Reviewer #2: Yes

2. Has the statistical analysis been performed appropriately and rigorously? 

Reviewer #1: Yes

Reviewer #2: No

3. Have the authors made all data underlying the findings in their manuscript fully available?

Reviewer #1: Yes

Reviewer #2: Yes

4. Is the manuscript presented in an intelligible fashion and written in standard English?

Reviewer #1: Yes

Reviewer #2: Yes

5. Review Comments to the Author

Reviewer #1: 1 . All the sections of the paper were well written.

2. The word count for the abstract was more than the prescribed number

3. Though the authors mentioned the limitation of the study but it will also be nice to include the strength of the study as well.

Reviewer #2: There is a disjoint between paragraph 3 and 4 . Paragraph 4 is a whole new concept and very unrelated to paragraph and was not properly introduced to assist the reader in understanding the concept.

The objective of the research was not SMART.

Data analysis was not described in clear terms as well as data presentation. The methods and tools used for data analysis were not described.

6. PLOS authors have the option to publish the peer review history of their article (what does this mean?). If published, this will include your full peer review and any attached files.

Reviewer #1: No

Reviewer #2: No

---

## [Author Response · Author response to Decision Letter 0]

28 Aug 2022

Reference: PONE-D-22-12589

Title: Bibliometric Analysis of Hotspots and Frontiers in Cancer-Related Fatigue Among Ovarian Cancer Survivors

Journal title: PLOS ONE

Authors: Yuanxia Liu, Qianxia Liu, Xiaolian Jiang* 

Dear Editor Prof. Kehinde Sharafadeen Okunade and dear reviewers,

We would like to give sincere gratitude to the editor for his kind letter and the reviewers for their constructive comments on our article. These positive comments and valuable suggestions are extremely important for the improvement of our article. All authors are doing their best to revise the manuscript to meet the requirements of your journals. 

In the rest of this letter, we will discuss your comments one by one and respond to each of them accordingly. A general summarization of all changes made to the article will also be provided. The changes we have made to the manuscript are highlighted in red text in the Revised Manuscript with Track Changes for your convenience.

1 Reply to comments one by one

1.1 Reviewer #1

Comment 1. All the sections of the paper were well written.

Respond 1. We are happy to hear you say that and thank you for your approval of our manuscript.

Comment 2. The word count for the abstract was more than the prescribed number

Respond 2. Thank you for your attention and patience. We have reduced the number of words in the abstract from the original 315 to 245, and these changes are all reflected in lines 15 to 54 of the Revised Manuscript with Track Changes (Figure 1).

Figure 1

Comment 3. Though the authors mentioned the limitation of the study but it will also be nice to include the strength of the study as well.

Respond 3. Thank you for your suggestion. We have added the strengths of this study to the article. These modifications can be seen in lines 453 to 458 of the Revised Manuscript with Track Changes (Figure 2).

Figure 2

1.2 Reviewer #2

Comment 1. There is a disjoint between paragraph 3 and 4. Paragraph 4 is a whole new concept and very unrelated to paragraph and was not properly introduced to assist the reader in understanding the concept.

Respond 1. Thank you for your suggestion. The writing between paragraph 3 and 4 is incoherent. The corrections are as follows: 

a) Paragraph 3 has been rewritten; 

b) Since the new concept mentioned in paragraph 4 is used in CiteSpace for results description and data analysis, this paragraph has been rewritten and placed in Data Analysis. 

These modifications can be seen in lines 91 to 119 of the Revised Manuscript with Track Changes (Figure 3). 

Figure 3

Comment 2. The objective of the research was not SMART.

Respond 2. Thank you for your precious suggestion. The specific objective of this study was to analyze the literature related to CRF in ovarian cancer patients published in WOSCC from 1991 - 2021 to understand the current state of research in the field and to provide evidence-based, visual evidence for scholars and policymakers. For this purpose, we used the bibliometric tool CiteSpace to measure the retrieved literature in terms of publication years, research countries/regions, institutions, authors, and journals to describe the current status and frontiers of research in the field. This article, finalized in May 2022, was written after a comprehensive search that included all literature in the field published in WOSCC up to December 31, 2021. We also provide further clarification in the article based on the reviewer's comment. These modifications can be seen in lines 120 to 126 of the Revised Manuscript with Track Changes (Figure 4).

Figure 4

Comment 3. Data analysis was not described in clear terms as well as data presentation. The methods and tools used for data analysis were not described.

Respond 3. Thank you for your suggestion. The corrections are as follows:

a) Because this paper uses the visual analysis tool Citespace, the data for countries/regions, institutions, and authors are mainly presented in graphical form (Figire2-7). Take Figure 2 as an example. The information in the upper left corner of the image describes the data including the software and version used, the date of the drawing, the timespan of the retrieved literature, the algorithm applied, network nodes, lines, and density. The image is also described appropriately in the Results of this paper. These descriptions can be seen on page 11 of lines 198-207, and 735-737 (Figure 5).

Figure 5

b) Data analysis has been added (Figure 6).

c) The methods and tools used for data analysis have been described (Figure 6).

These modifications can be seen in lines 144 to 176 of the Revised Manuscript with Track Changes.

Figure 6

2 Summarize all the modifications in the article

2.1 Page 2 of lines 15, 18-20, 23-30, 32-33 

2.2 Page 3 of lines 34-42, 48, 50-54

2.3 Page 5 of lines 63

2.4 Page 6 of lines 85-86, 88, 91-101

2.5 Page 7 of lines 102-123

2.6 Page 8 of lines 139, 144-145

2.7 Page 9 of lines 146-167

2.8 Page 10 of lines 168-176

2.9 Page 14 of lines 242-244 

2.10 Page 16 of lines 278 

2.11 Page 25 of lines 433 

2.12 Page 26 of lines 448, 453-458 

2.13 Page 27 of lines 475-481 

2.14 The reference format has been modified concerning journals that have been published in PLOS ONE. These modifications can be seen on Pages 28 to 33.

2.15 We have uploaded figures to the PACE digital diagnostic tool and revised the figures in the article to meet PLOS requirements. 

2.16 There was an error in one author's affiliation, which has been corrected in the new manuscript.

We are not sure if our modifications and explanations fully answer your question. If you need us to make any further modifications, please send us an email and we are willing to make further changes until our article meets the publication requirements of your journal. Thank you again for your time and interest. We look forward to hearing from you.

Kind regards,

The Authors.

（All figures can be seen in Response to Reviewers)

---

## [Editor Report · Decision Letter 1]

4 Sep 2022

Bibliometric Analysis of Hotspots and Frontiers in Cancer-Related Fatigue Among Ovarian Cancer Survivors

PONE-D-22-12589R1

Dear Dr. Jiang,

We’re pleased to inform you that your manuscript has been judged scientifically suitable for publication and will be formally accepted for publication once it meets all outstanding technical requirements.

Kind regards,

Kehinde Sharafadeen Okunade

Academic Editor

PLOS ONE
---

## [Editor Report · Acceptance letter]

14 Sep 2022

PONE-D-22-12589R1 

Bibliometric Analysis of Hotspots and Frontiers in Cancer-Related Fatigue Among Ovarian Cancer Survivors 

Dear Dr. Jiang:

I'm pleased to inform you that your manuscript has been deemed suitable for publication in PLOS ONE. Congratulations! Your manuscript is now with our production department. 

Kind regards, 

on behalf of

Dr. Kehinde Sharafadeen Okunade 

Academic Editor

PLOS ONE